# Intracranial Flow Volume Estimation in Patients with Internal Carotid Artery Occlusion

**DOI:** 10.3390/diagnostics12030766

**Published:** 2022-03-21

**Authors:** Piotr Kaszczewski, Michał Elwertowski, Jerzy Leszczyński, Tomasz Ostrowski, Joanna Kaszczewska, Zbigniew Gałązka

**Affiliations:** Department of General, Endocrine and Vascular Surgery, Medical University of Warsaw, Banacha 1a, 02-097 Warszawa, Poland; tostr@vp.pl (T.O.); joanna.kaszczewska@gmail.com (J.K.); zbigniew.galazka@wum.edu.pl (Z.G.)

**Keywords:** Doppler ultrasonography, carotid occlusion, carotid artery disease, cerebral blood flow, cerebrovascular reserve

## Abstract

(1) Background: Carotid artery occlusion (CAO) in population studies has a reported prevalence of about 6 per 100,000 people; however, the data may be underestimated. CAO carries a significant risk of stroke. Up to 15% of large artery infractions may be secondary to the CAO, and in 27–38% of patients, ischaemic stroke is a first presentation of the disease. The presence of sufficient and well-developed collateral circulation has a protective influence, being a good prognostic factor in patients with carotid artery disease, both chronic and acute. Understanding the mechanisms and role of collateral circulation may be very important in the risk stratification of such patients. (2) Materials and Methods: This study included 46 patients (mean age: 70.5 ± 6 years old; 15 female, mean age 68.5 ± 3.8 years old and 31 male, mean age 71.5 ± 6.7 years old) with unilateral or bilateral ICA occlusion. In all patients, a Doppler ultrasound (DUS) examination, measuring blood flow volume in the internal carotid artery (ICA), external carotid artery (ECA), and vertebral artery (VA), was performed. The cerebral blood flow (CBF) was compared to the previously reported CBF values in the healthy population >65 years old. (3) Results: In comparison with CBF values in the healthy population, three subgroups with CBF changes were identified among patients with ICA occlusion: patients with significant volumetric flow compensation (CBF higher than average + standard deviation for healthy population of the same age), patients with flow similar to the healthy population (average ± standard deviation), and patients without compensation (CBF lower than the average-standard deviation for healthy population). The percentage of patients with significant volumetric flow compensation tend to rise with increasing age, while a simultaneous decline was observed in the group without compensation. The percentage of patients with flow similar to the healthy population remained relatively unchanged. ICA played the most important role in volumetric flow compensation in patients with CAO; however, the relative increase in flow in the ICA was smaller than that in the ECA and VA. Compensatory increased flow was observed in about 50% of all patent extracranial arteries and was more frequently observed in ipsilateral vessels than in contralateral ones, in both the ECA and the VA. In patients with CAO, there was no decrease in CBF, ICA, ECA, and VA flow volume with increasing age. (4) Conclusions: Volumetric flow compensation may play an important predictive role in patients with CAO.

## 1. Introduction

Atherosclerosis is currently considered one of the most serious health burdens. Its prevalence increases significantly with age, affecting about 20% of people under 60 years old, and more than 50% of those over 85 [1].

Stroke, affecting about 15 million people worldwide annually, is one of the leading causes of mortality. One-third of these patients die, and 15% remain permanently disabled [2].

It is estimated that 85% of all strokes are of ischaemic origin, and 15–20% of them are secondary to haemodynamically significant atherosclerotic lesions of the bifurcation of common carotid artery (CCA) and the proximal part of the internal carotid artery (ICA) [3,4,5,6].

Two main pathogenetic mechanisms of ischaemic stroke have been identified: haemodynamic impairment due to malperfusion secondary to the significant stenosis. and thromboembolism from a vulnerable atherosclerotic plaque [7].

About 10% of all strokes, and up to 60% of infarcts in patients with carotid artery disease, are watershed (WS) or border-zone infarcts. They appear in the borders of the vascular beds of the main brain-supplying arteries: the anterior cerebral artery (ACA), the posterior cerebral artery (PCA), and the middle cerebral artery (MCA) [8].

Historically, they were thought to be caused mainly by hypoperfusion; however, nowadays it is believed that haemodynamic impairment and microembolisation may act synergistically [7].

The presence of sufficient and well-developed collateral circulation has a protective influence, being a good prognostic factor in patients with carotid artery disease, both chronic and acute. Understanding the mechanisms and role of collateral circulation may be very important for the risk stratification of such patients [9].

It was recently observed that, among the group of asymptomatic patients with ICA stenosis and occlusion, there are patients with cerebral blood flow volume (CBF) changes in comparison to the healthy population of the same age. Three subgroups have been identified: patients with elevated cerebral blood flow volume; those with undisturbed flow volume, similar to the healthy population; and those with diminished flow volume [5,10,11].

The changes in blood flow volume and the level of compensation may be related to the presence of neurological symptoms [12,13].

The increased CBF, secondary to the presence of significant volumetric flow compensation, has a protective effect against developing ischaemic symptoms, including TIA or stroke. Asymptomatic patients tend to have higher CBF than symptomatic ones, with similar ICA narrowing. The chance of observing ischaemic symptoms in patients with ≥70% ICA stenosis and diminished CBF is twice that in those with a similar ICA stenosis degree and significantly increased flow volume [11].

The importance of collateral circulation is clear in patients with ICA agenesis, aplasia, or hypoplasia. The vast majority of such patients are asymptomatic due to well-developed collateral circulation via either CoW or the persistent embryonical vessels and collaterals. One of the collateral pathways is ECA-ICA anastomosis, when the intracranial arteries are supplied by the branches of ascending pharyngeal and internal maxillary arteries [12].

The assessment of cerebral inflow in Doppler ultrasonography may provide a novel and easily accessible tool of identifying patients prone to cerebral ischaemia [11]. The multivessel character of compensation, with enhanced role of ECA, justifies the importance of including this artery in the estimation of CBF.

Carotid artery occlusion (CAO) in population studies has a reported prevalence of about 6 per 100,000 people; however, the data may be underestimated. CAO carries a significant risk of stroke. Up to 15% of large artery infarctions may be secondary to the CAO, and in 27–38% of patients significant, massive ischaemic stroke is the first presentation of the disease [14,15,16,17].

Patients diagnosed with the CAO also have a higher (by 2–5.5%) annual risk of stroke ipsilateral to the occlusion. This risk increases when CAO is accompanied by asymptomatic contralateral ICA stenosis—the combined, annual risk of stroke is then estimated for up to 10%. Perioperative mortality as well as the periprocedural risk of stroke is also increased in this group of patients; it is reported for 1.8–3.7% for conventional procedures and 3.3–7% for stenting. Following the CAO diagnosis, the five-year mortality rate is as high as 30–40% [15,18,19,20].

The aim of this study was to assess the compensatory mechanisms with Doppler ultrasonography, examining the circulation pathways and flow volume in patent extracranial arteries, in the group of patients over 65 with internal carotid artery occlusion.

## 2. Materials and Methods

This study included 46 patients (mean age of group: 70.5 ± 6 years old; 15 female, mean age 68.5 ± 3.8 years old and 31 male, mean age 71.5 ± 6.7 years old) with ICA occlusion. The details are presented in Table 1.

The flow volume values, which in this study are hereafter called “reference” values or the “proposed reference standard,” were established by our team based on a group of 123 healthy volunteers (without concomitant disorders, which could influence CBF values) and published in 2020. The reference range is presented as the average flow volume (mL/min) ± standard deviation (mL/min); see Table 2 [5]. 

In this study, the flow volumes in extracranial arteries (CCA, ECA, ICA, and VA) in patients with ICA occlusion were compared with those obtained for a group of healthy volunteers in order to determine the pathways and the degree of compensatory circulation.

In this study, values exceeding the proposed reference values (see Table 2) for average + standard deviation are referred as to having “compensatory increased flow” or “significant compensation.”

The values within the proposed reference are referred to as “Flow volume within reference value” or “mild compensation”—in the presence of a major reduction in flow in one of the carotid arteries, an increase in the other vessels allows it to maintain the CBF within the proposed standards.

A blood flow volume lower than the proposed reference value for average and standard deviation was referred as to “no compensation” or “decreased flow volume.”

Recruitment of patients to both groups in terms of concomitant disorders was conducted according to the previously described protocol, in order to eliminate its influence on cerebral blood flow volume, except for ICA stenosis/occlusion [7].

Before the examination, informed consent was given by all study participants. The study was given the approval of the Medical University of Warsaw Bioethical Committee.

In all patients, a DUS examination measuring blood flow volume in the common carotid arteries (CCAs), patent internal carotid artery (ICA), external carotid arteries (ECAs), and vertebral arteries (VAs) was performed. The flow volume in the CCA was measured as a control; the measurements were considered accurate when the sum of the flow volumes in the ipsilateral ICA and ECA (measured distally to the superior thyroid artery), or ECA only (in the case of ipsilateral ICA obstruction), was slightly lower than the flow volume in the CCA (due to flow volume loss in the superior thyroid artery). 

The cerebral blood flow (CBF) was calculated as the grand total of the flow volumes in all the aforementioned patent extracranial arteries: ICA, ECAs—distal to the origin of the superior thyroid artery, and VAs. 

Examinations were conducted following a previously described protocol [5], by the same experienced sonographer, using a Canon Aplio i800 ultrasound scanner with Linear i11LX3 transducer (Canon Medical Systems Corporation, Otawara, Tochigi, Japan). The blood flow volumes were calculated using the ultrasound scanner’s semiautomatic program. 

The diameter of each vessel was measured using three different techniques: B-mode, SMI (superb microvascular imaging) mode and B-mode combined with SMI image. All measurements (both diameter measurement and volumetric assessment) were carried out three times and their average was considered the final result.

### Statistical Analysis

Statistical analysis was performed with Statistica 13 (StatSoft Polska Sp. z o.o., Krakow, Poland).

For the comparison of the two groups, the *t*-test and the Mann–Whitney U test were used. The Shapiro–Wilk test was performed as a test of normality. Levene’s test was used to assess the equality of variances. The normal distribution of data with equal variances was a prerequisite to use the *t*-test. With no equality of variances, the *t*-test with Cochran–Cox correction was performed. When one of the variables had abnormal distribution, a nonparametric Mann–Whitney U test was performed. The results were considered statistically significant when the *p*-value was below 0.05. Additionally, a Bonferroni correction for multiple comparisons was performed in order to reduce the risk of type I errors; the significance level was set to 0.005.

Additionally, a linear regression analysis was performed. The correlation was considered statistically significant when the *p*-value was below 0.05.

## 3. Results

The contents of this section are as follows:Section 3.1—the number and percentage of patients with total CBF changes in different age groups.Section 3.2—the differences between the flow volume in the arteries with “significant compensation” and the reference values.Section 3.3—the pathways of collateral circulation—the number and percentage of arteries with compensatory increased flow volume, as well as side-by-side differences (contralateral vs. ipsilateral vessels).Section 3.4—Correlation of CBF and flow volumes in extracranial arteries with age.Section 3.5—Comparison of the number of male and female participants in our study group, by increasing age.

### 3.1. Cerebral Blood Flow Volume in the Whole Study Group (46 Patients)

Increased flow volume was observed in 13/46 patients (28.3%). Flow volume similar to healthy volunteers was observed in 20/46 patients (43.4%). Decreased flow volume was observed in 13/46 patients (28.3%).

The percentage of patients with significant volumetric flow compensation tended to rise with increasing age (13.5%, 65–69 years old; 33.5%, 70–74; 71.4%, >80), while a simultaneous decline was observed in the group without compensation (45.4%, 65–69; 6.6%, 75–79; 0%, >80). In the group aged ≥80, no patients with diminished blood flow volume were observed.

The percentage of patients with flow similar to the healthy population remained relatively unchanged (41%, 65–69; 60%, 70–74; 28.5%, >80).

Detailed data concerning the flow compensation in the whole study group are presented in Table 3 and Figure 1.

### 3.2. The Degree of Compensation in Extracranial Arteries with Compensatory Increased Flow

The level of compensation rose with age in the ICA—from 149.3% (65–69 years old), to 170.2% (70–74), to 199.8% (≥80). 

In the ECA, level of compensation remains relatively unchanged (216.9%, 203.6%, 222.5%).

In VA, a slight decrease in observed in the 70–74 age group (from 279% to 246.7%); however, a prominent increase in the level of compensation is observed in patients ≥80 (386.9%).

The data concerning the level of compensation in all extracranial arteries with increased flow volume are presented in Table 4 and Figure 2.

### 3.3. The Pathways of Volumetric Flow Compensation in the Extracranial Arteries

The data concerning pathways of collateral circulation in patients with unilateral ICA occlusion are presented in Table 5, and with bilateral ICA occlusion in Table 6.

Volumetric flow compensation in patients with unilateral ICA occlusion was most frequently observed in:23 contralateral ICA16 contralateral ECA17 contralateral VA23 ipsilateral ECA22 ipsilateral VA.

In patients with bilateral ICA occlusion, flow compensation was observed in all (8/8) VA, and in 6/8 ECA.

Altogether, in the whole study group, the compensatory increased flow volume was around 50% of all ICAs, ECAs, and VAs. The detailed percentages of the ICA, ECA, and VA with flow compensation are presented in Figure 3.

The compensatory increased flow was more frequently observed in ipsilateral vessels than in contralateral ones—see Figure 4.

### 3.4. The Correlation between Volumetric Flow Compensation, Cerebral Blood Flow Volume, and Age

In the study group, there was no significant correlation observed between age and the degree of compensation in ICA, ECA, VA and CBF. The data are presented in Figure 5, Figure 6, Figure 7 and Figure 8.

### 3.5. Number of Male and Female Patients in Study Group

In our study group, we observed a continuous decrease in the percentage of female patients. No female patients were observed in the oldest age group (≥80)—see Figure 9.

## 4. Discussion

Significant differences in the annual incidence of stroke were observed between symptomatic (up to 27%) and asymptomatic patients (up to 5 %). Such variability of the disease manifestation may be explained by the development and effectiveness of the collateral circulation [10,21].

Compensatory mechanisms bring about flow changes in all extracranial arteries. It is believed that the primary compensatory reaction for the CAO is flow increase in the contralateral ICA [22].

When the collateral circulation is poorly developed, the risk of occurrence of neurological symptoms rises, especially in low-flow states [23].

Fang et al. examined 586 acute ischaemic stroke patients (mean age, 67.5 ± 12.4), identifying 81 with unilateral ICA stenosis and 31 with bilateral ICA stenosis in the carotid duplex. They discovered, by assessing the blood flow volume with Doppler ultrasonography, that in patients with unilateral ICA stenosis, the blood flow volume (BFV) in contralateral CCA was significantly higher than in ipsilateral CCA (325.5 ± 99.8 mL/min vs. 242.2 ± 112.2 mL/min). Among patients with bilateral ICA stenosis, the authors also observed flow changes in vertebral arteries: BFV in bilateral VAs increased significantly compared with patients without ICA stenosis (146.2 ± 70.4 mL/min vs. 109.1 ± 52.5 mL/min), accounting for almost 22% of the whole brain BFV, which was significantly higher than 14.8% in those without ICA stenosis. The authors concluded that, in patients with unilateral carotid stenosis, contralateral carotid blood flow increases to compensate for the decreased blood flow, while posterior circulation may compensate for the decreased brain perfusion in those with bilateral carotid stenosis [24].

Sundaram et al. demonstrated that assessment of collateral circulation in CTA is an important tool for predicting three-month outcome in patients with symptomatic ICA occlusion. The authors stressed the role of secondary collaterals (leptomeningeal collaterals and ophthalmic artery), which might be more influential than Willisian flow in determining the outcome in such patients [25].

Liu et al. examined the correlation between CoW morphology and collateral pathways in 30 patients with bilateral carotid occlusion. They concluded that, with complete CoW, the blood flow compensation to the ischaemic area is mainly provided through primary collateral circulation, but with incomplete CoW, secondary collateral circulation plays the most important role [26].

Zarrinkoob et al. underlined the need for simultaneous investigation of the haemodynamics of the entire cerebral arterial tree. Examining patients with ICA stenosis, using four-dimensional phase-contrast magnetic resonance imaging, they found that contralateral internal carotid artery supplied the bilateral anterior cerebral artery territory. The authors stated that compromised blood flow in MCA is not necessarily related to the degree of carotid stenosis [27].

ECA and VA are believed to play a secondary role in compensation in patients with CAO. Nicolau et al. performed a study concerning changes in the VA blood flow in patients with ICA occlusion in comparison to healthy volunteers. In patients with CAO, the authors detected increased flow velocities and blood flow volumes in VA. More pronounced side-by-side differences in flow velocity and volume were also observed in patients with CAO. The authors also observed a mean increase of 17.6% in VA flow volume in comparison to healthy volunteers (150.16 ± 77.92 versus 123.68 ± 52.38 mL/min). A greater increase in flow volume was observed in the ipsilateral VA than on the contralateral side (83.84 ± 66.31 versus 66.31 ± 45.40 mL/min). Similar tendencies were observed concerning the flow velocity, which was higher on the ipsilateral side (57.67 ± 23.73 versus 47.8 ± 18.49 cm/s) [28].

Van Laar et al. investigated the role of ECA in the compensatory circulation. In the 30 patients with unilateral ICA occlusion, they identified three groups. In 20%, ECA did not contribute to the brain perfusion. In another 20%, the ECA supplied the focal region of the ipsilateral medial cerebral artery (MCA) territory, while in 60% ECA supplied MCA and part of the ipsilateral anterior cerebral artery (ACA) territory. The authors concluded that focal brain lesions may strongly depend on the ECA blood supply, even in patients with limited ECA collateral flow [29].

The role of the ECA in the compensatory circulation in the ICA occlusion is important. While in physiological conditions ECA provides very little or no blood supply to the central nervous system, in patients with severe ICA stenosis or occlusion it becomes a vital collateral pathway. In cases of bilateral ICA occlusion, the ECA may be responsible for up to one-third of CBF [30].

In patients with severe bilateral atherosclerotic lesions of the ICA and ECA, ECA revascularization may result in a significant (15–39% ipsilateral and 12–52% contralateral) cerebral blood flow volume increase [31].

It was also reported that ECA endarterectomy in patients with ICA occlusion may result in a reduction of neurological symptoms, such as amaurosis fugax, hemispheric TIA, and nonlateralizing symptoms [32].

In this study, the pathways of collateral circulation were studied. While it is generally considered that ICA plays the most important role in the collateral circulation, in our study this was only partially confirmed. Consistent with other, previously published studies, the flow excess in the ICA was found to be the highest in all age groups. However, compensatory increased flow in the contralateral ICA was present in only 50% of our patients. Taking into account other extracranial vessels, compensation was more frequently observed in ipsilateral vessels (23 ECA, 22 VA vs. 16 ECA, 17 VA). If we look at the percentage, volumetric flow compensation was present in about 50% of all patent ICAs (54.8%), ECAs (48.9%), and VAs (51.1%). However, the number of patent ECAs and VAs was twice that of patent ICAs, which emphasizes the importance of these vessels as pathways of collateral circulation.

However, it must be stressed that, while the flow increase in the ICA was highest in terms of absolute numbers, relatively it was the lowest. The relative flow increases in the ICA ranged from 150% to almost 200% in comparison to healthy volunteers. The relative increase in flow in the ECA and VA was much more prominent—204–223% and 247–322%. This clearly shows that, while in the ICA the flow volume can nearly double, in the VA it can more than triple. The ability of VA to increase flow volume is surprisingly high, surpassing previously reported values [24,28].

The percentages are based on average values from whole study groups; in individual cases, the compensation pattern may vary (see Figure 10 and Figure 11).

Discussing the role of the ECA, we are conscious of the need for careful interpretation of the volumetric flow compensation. Collateral circulation is based on the intracranial and extracranial anastomoses. The former includes vertebrobasilar circulation, the circle of Willis (CoW), the tectal plexus, the anterior, middle, and posterior cerebral arteries branch, as well as leptomeningeal anastomoses. The extracranial–intracranial anastomoses cover three main regions: the orbital region (via the ophthalmic artery connecting the internal maxillary and internal carotid vascular beds), the petrous–cavernous region (via the inferolateral trunk, the petrous branches of the internal carotid artery, and the meningohypophyseal trunk to the carotid artery), and the upper cervical region (via the ascending pharyngeal, the occipital, and the ascending and deep cervical arteries to the vertebral artery) [33,34].

Using only ultrasonography, we are not able to determine what part of the flow volume excess in the ECA supplies the central nervous system; however, the literature data, together with our results, allow us to claim that ECA is a vital collateral pathway. We have to bear in mind that there are also several other extracranial–intracranial anastomotic pathways, which cannot be assessed in DUS.

The circle of Willis is considered the most important arterial anastomosis. It was described nearly 400 years ago by Thomas Willis, who also described its role as a compensatory mechanism in the case of occlusion or stenosis of the ICA or VA [35].

Iqbal has analysed the anatomy of the circle of Willis. In his work, the normal configuration of the CoW, without abnormalities, was identified in only 48% of cases. In the literature, physiological configuration of the CoW is identified in 14.2–72.2% of patients [36,37,38].

The most common anomaly is the presence of hypoplasia of one of the vessels of the CoW (in the study by Iqbal: posterior communicating artery, 10%; posterior cerebral artery (circular part P-1 segment), 6%; anterior cerebral artery (circular part A-1 segment), 4%; and anterior communicating artery, 4%). The accessory vessels were observed in 12% of cases, mainly in the anterior portion of the CoW: anterior communicating (8%) and anterior cerebral arteries (4%). In 10% of cases, the anomalous origin of the arteries (most commonly posterior cerebral artery originating from ICA—10%) was observed. Absence of the vessels was observed least commonly, in the posterior communicating artery (6%) [36].

The role of the CoW in the development of TIA or stroke seems obvious; however, there are contradictory opinions. Studies by Yu-Ming et al., Shahan, Seeters et al., the SMART study group, Eldrea et al., and Badacz et al. demonstrated a significant correlation between abnormalities in the CoW and ischaemic stroke [39,40,41,42,43,44].

The meta-analysis by Oumer et al. performed with the 2718 participants did not demonstrate a statistically significant correlation, but a nonsignificant positive association between variation in COW and ischaemic stroke (pooled OR: 1.38 (95% CI 0.87, 2.19)) was identified [45].

Jongen et al. noted a gradual decrease in cerebral blood flow in patients with increasing severity of carotid stenosis, independent of the CoW morphology. They concluded that collateral pathways, including ophthalmic and leptomeningeal vessels, may compensate for the CoW collaterals [46].

Vrselja et al., in their analysis of the function of the CoW, concluded that the compensatory function of the circle of Willis may be incorrect. They pointed out that the communicating arteries are too small or hypoplastic in the majority of the population to facilitate effective blood transfer. The authors claimed that CoW may serve as a passive energy (pressure)-dissipating system, transferring pressure without considerable blood flow from the high-pressure end to the low-pressure end, where the pulse wave and blood flow arrive asynchronously [35].

Myrcha et al. examined the factors influencing cross-clamping intolerance during carotid endarterectomy. They suggested that the anatomy of the circle of Willis itself is not a strong predictive factor for the prognosis of cross-clamping intolerance. The combined examination, including CoW anatomy together with brain perfusion tests, facilitates the evaluation of cross-clamping intolerance risk with a much higher probability [47].

Ultrasound assessment of cerebral blood flow volume in extracranial arteries may provide indirect insight into brain perfusion. If the blood flow volume in the extracranial arteries is high, the CBF should also be high, which implies that the compensatory mechanisms are sufficient. Further studies examining brain perfusion and correlating it with CBF assessment through Doppler ultrasonography are needed to confirm this hypothesis.

A fact that was also surprising was the lack of correlation between age and volumetric flow compensation. In our previous study, in the group of healthy volunteers, we noted a gradual decrease in cerebral blood flow volume, which was the result of a significant decrease in flow volume, mainly in the ICA and, to a lesser extent, in the ECA. This was accompanied by ICA PSV, EDV, and ECA PSV decreases. The cerebral blood flow volume decline was more rapid in elderly people than in younger age groups, accelerating rapidly after age 70 [5]. In patients with carotid artery occlusion, we did not observe the same tendencies and trends as those, which were seen in the healthy population. In patients with CAO, CBF remained mostly unchanged with increasing age. A slight decreasing tendency in flow volume was observed in ECA and VA, while an increasing trend was noted in ICA. We must be careful when looking for an explanation for this observation. What is worth noticing is the fact that, in the oldest age group, ≥80, there were no patients without flow compensation. The percentage of patients with flow compensation gradually increased with age (excluding the group aged 75–79, which is not representative). This was accompanied by a simultaneous decrease in the percentage of patients without flow compensation. The explanation that is most logical must, however, be confirmed in studies with a much larger groups of patients: volumetric flow compensation promotes and facilitates patient survival. This may also explain the fact that, in the group of patients with carotid occlusion, there were no physiological correlations of decreasing blood flow, e.g., CBF and flow in extracranial arteries, which were observed in the healthy population.

In our study, one surprising observation has been made: the percentage of female patients with CAO decreased much more rapidly than that of males. We do not want to draw any firm conclusion based on that observation, though; this striking tendency must be verified in a study on a much larger population. We can only say that we analysed the patients under the supervision of our department, which is representative of a regional population, and observed such a tendency.

Our study had several limitations. With this study design, based on ultrasonography of extracranial arteries, we were not able to determine the correlation of flow compensation with the anatomy of intracranial arteries, nor to examine flow in those vessels. In our study, the CBF values were measured for a certain time—we did not have the ability to conduct a longitudinal study, assessing and monitoring the flow dynamics over time, before and after the occlusion. This may be an interesting topic for future research. Our method allows for the identification of patients with and without flow compensation. It is mainly intended for clinicians, so its greatest advantages are simplicity, availability, and reproducibility—such measurements can be performed with almost any ultrasound scanner by a skilled medical professional.

In order to introduce this method as a diagnostic tool, randomized, multicentre studies have to be conducted on representative groups of patients.

Intracerebral circulation in patients with severe (≥70%) ICA narrowing is subject to different flow compensation, with the degree of flow volume change varying even with a similar degree of stenosis. ICA occlusion provides an excellent pathophysiological model of flow impairment that is not affected by the originally disturbed vessel, thus accentuating other mechanisms providing proper brain perfusion. In our study, contralateral ICA was the most effective single vessel supplying surplus flow to underperfused regions of the brain. However, the combined additional flow in ECA + VA was more effective as a provider of supplementary brain perfusion.

The presence of collateral circulation is a factor of the utmost importance in promoting patient survival. Patients with well-developed collateral circulation, receiving the best medical therapy, have a much lower risk of ischaemic stroke than the group without collateralization (13.3% versus 6.3%, for disabling or fatal stroke; 27.8% versus 11.3% for hemispheric stroke) [48].

## 5. Conclusions

Volumetric flow compensation may play an important role in risk stratification of patients with carotid occlusion.

Volumetric flow compensation may be a factor influencing patient survival: in the study group, the increasing percentage of patients with significant volumetric flow compensation was observed with increasing age, while the opposite trend (a decreasing percentage of patients) was observed in the group without flow compensation.

ICA is the most important single vessel in volumetric flow compensation; however, the accentuated role of the ECA and VA highlights their impact, which might be more important than previously stated.

## Figures and Tables

**Figure 1 diagnostics-12-00766-f001:**
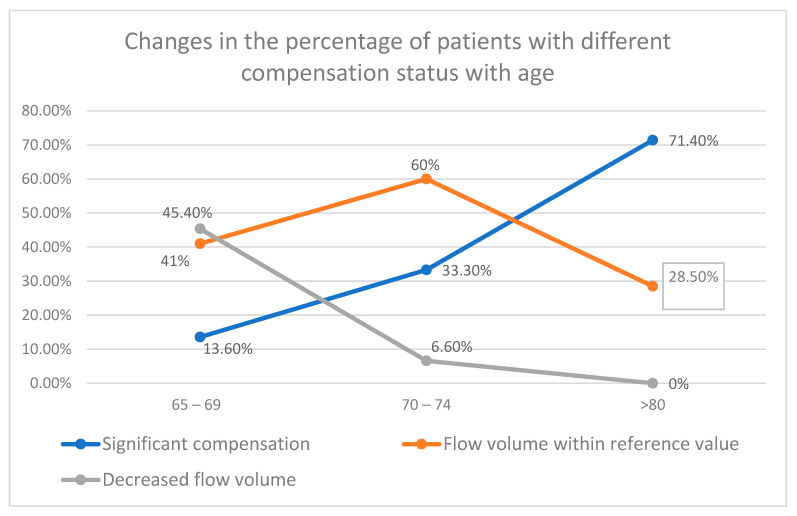
Tendencies in CBF changes in different age groups. The percentage of patients with significant flow compensation tends to rise with increasing age. The opposite trend is observed in the group with decreased flow volume. The age 75–79 group was excluded from the analysis.

**Figure 2 diagnostics-12-00766-f002:**
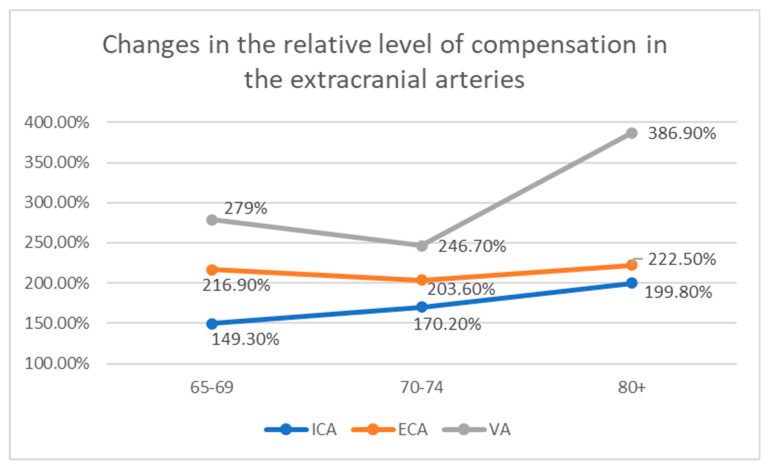
Changes in the relative level of compensation in extracranial arteries in patients with ICA occlusion. The relative level of compensation rises in ICA and VA with increasing age, while in ECA it remains almost unchanged.

**Figure 3 diagnostics-12-00766-f003:**
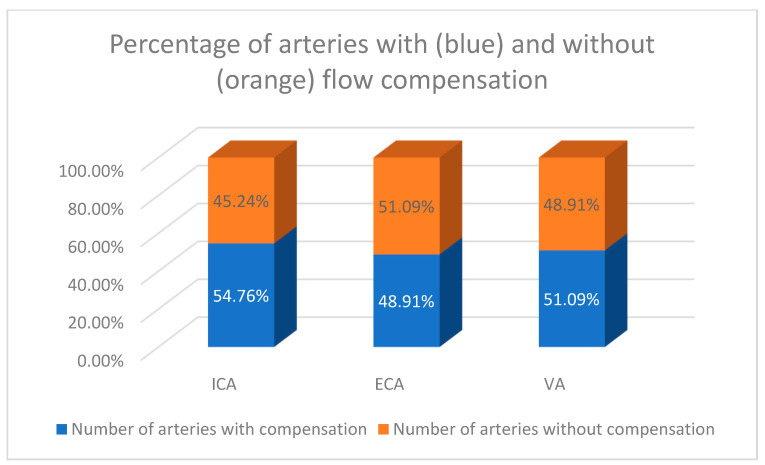
The percentage of patent arteries with and without flow compensation. The compensatory increased flow volume was most frequently observed in the contralateral ICA—54.76%, in 48.91% of ECAs, and in the 51.09% of VAs.

**Figure 4 diagnostics-12-00766-f004:**
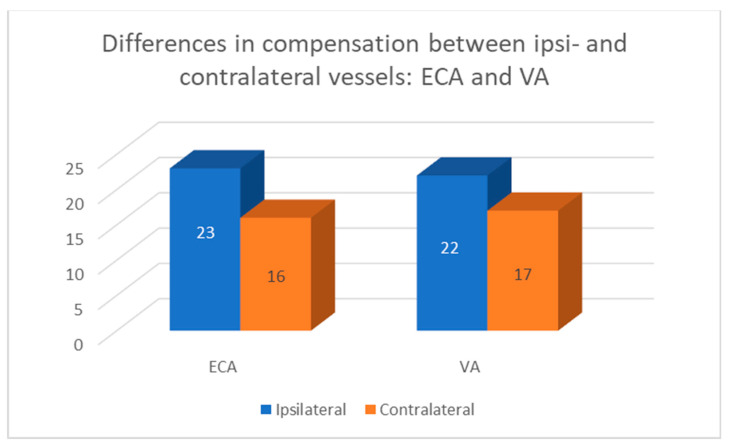
Differences in compensation between ipsilateral and contralateral vessels. Compensatory increased flow was more frequently observed in ipsilateral vessels (23 vs. 16—ECA) and 22 vs. 17—VA).

**Figure 5 diagnostics-12-00766-f005:**
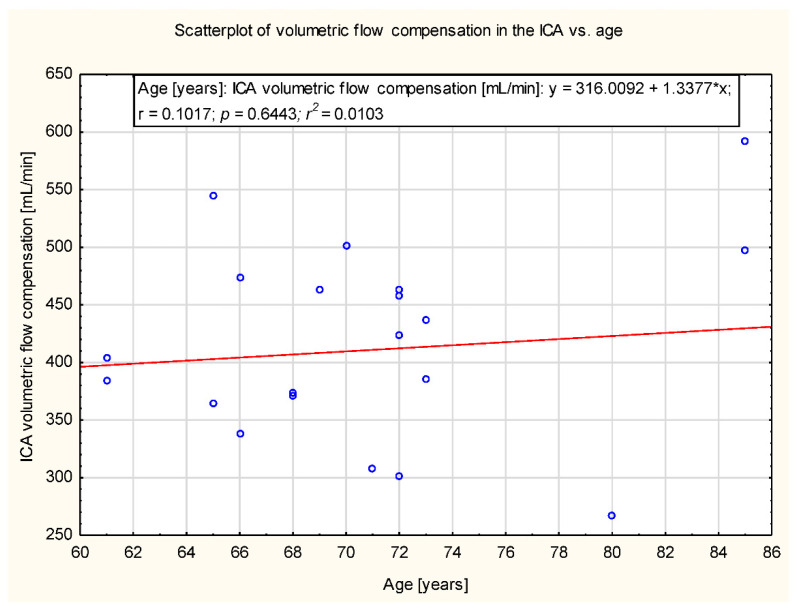
Scatter plot of the volumetric flow compensation in ICA and age (y = 316.0092 + 1.33778x). No correlation was observed (*p* = 0.64).

**Figure 6 diagnostics-12-00766-f006:**
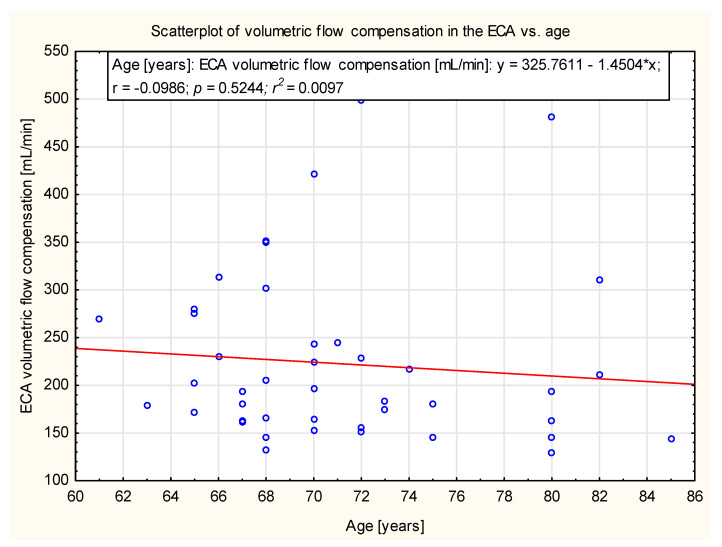
Scatter plot of the volumetric flow compensation in ECA and age (y = 325.7611 − 1.45048*x). No correlation was observed (*p* = 0.52).

**Figure 7 diagnostics-12-00766-f007:**
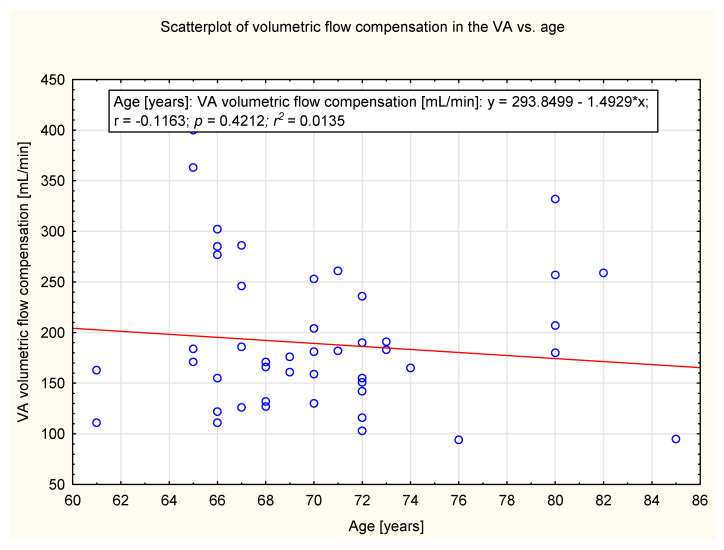
Scatter plot of the volumetric flow compensation in VA and age (y = 293.8499 − 1.4929*x). No correlation was observed (*p* = 0.42).

**Figure 8 diagnostics-12-00766-f008:**
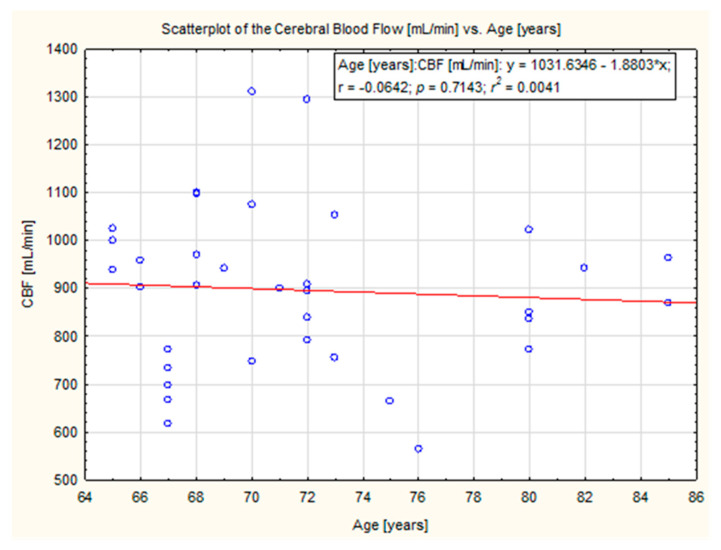
Scatter plot of cerebral blood flow volume and age (y = 1031.6346—1.8803*x). No correlation was observed (*p* = 0.71).

**Figure 9 diagnostics-12-00766-f009:**
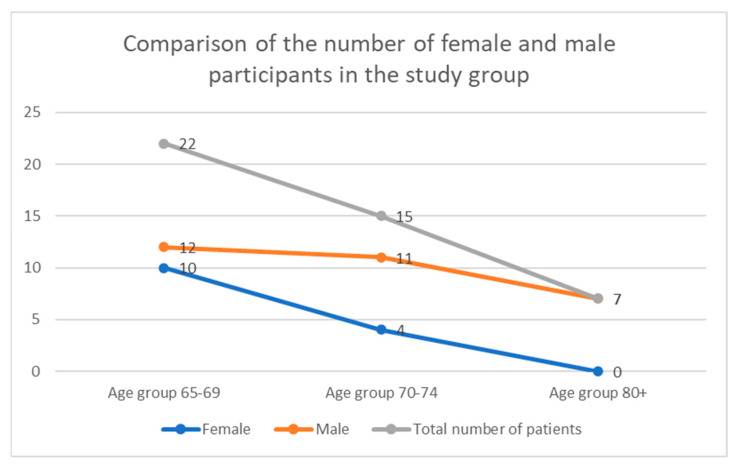
The number of female and male participants in our study group.

**Figure 10 diagnostics-12-00766-f010:**
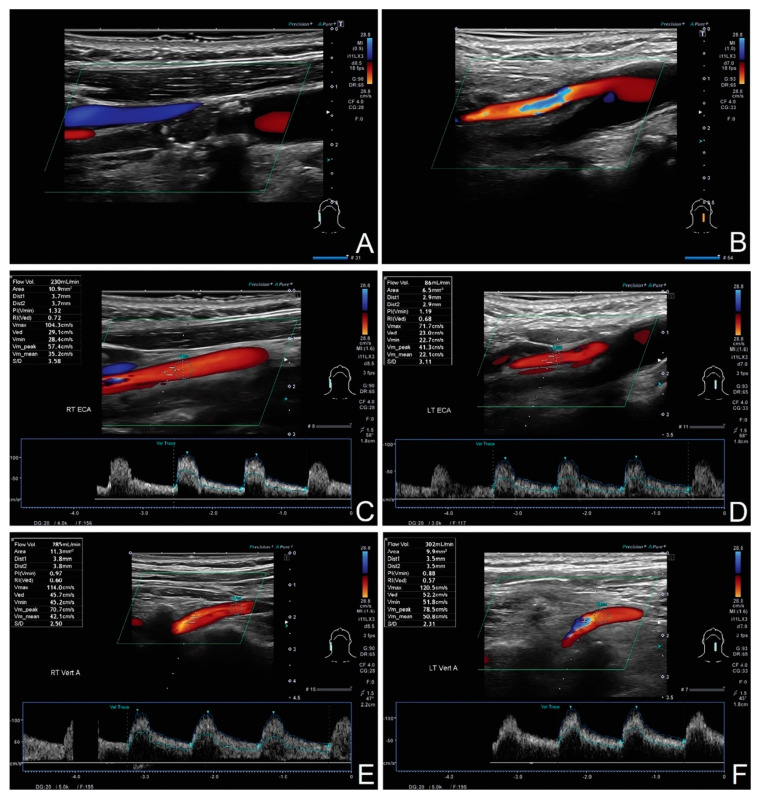
Compensation in patient with bilateral ICA occlusion. (**A**) Right-sided ICA occlusion; (**B**) left-sided ICA occlusion; (**C**) compensatory increased flow in RECA with flow volume of 230 mL/min; (**D**) flow volume in LECA of 86 mL/min; (**E**) compensatory increased flow in RVA with flow volume of 285 mL/min; (**F**) compensatory increased flow in LVA with flow volume of 302 mL/min. Flow was within the reference values despite bilateral CAO.

**Figure 11 diagnostics-12-00766-f011:**
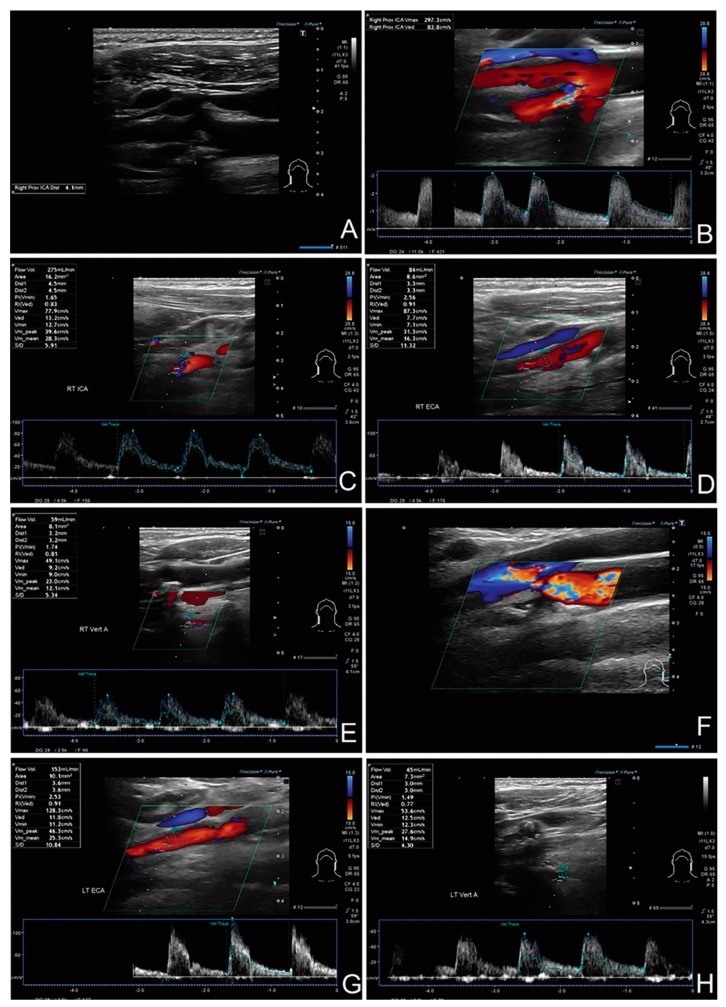
Patient with left-sided ICA occlusion, 70% right-sided ICA stenosis, and no significant flow compensation in other extracranial arteries. (**A**,**B**) About 70% RICA stenosis with flow velocity of 2.97/0.83 m/s; (**C**) distal part of the ICA with flow volume of 275 mL/min; (**D**) flow volume in RECA of 84 mL/min—no compensation; (**E**) flow volume in RVA of 59 mL/min—no compensation; (**F**) left-sided LICA occlusion; (**G**) slightly compensatory increased flow in LECA with flow volume of 153 mL/min; (**H**) flow volume in LVA of 65 mL/min—no compensation. Due to the lack of volumetric flow compensation, early elective surgical treatment should be considered.

**Table 1 diagnostics-12-00766-t001:** Study group characteristics.

	Female	Male	Total
Number of patients	15	31	46
Mean age ± std. dev.	68.5 ± 3.8 years old	71.5 ± 6.7 years old	70.5 ± 6 years old
Number of isolated LICA occlusions	2	16	18
Number of isolated RICA occlusions	10	14	24
Bilateral ICA occlusion	3	1	4
Age: 65–69	10	12	22
Age: 70–74	4	11	15
Age: 75–79	1	1	2
Age: 80+	0	7	7

**Table 2 diagnostics-12-00766-t002:** Reference flow volume in extracranial arteries RICA, LICA—right and left internal carotid artery. RECA, LECA—right and left external carotid artery. RVA, LVA—right and left vertebral artery. Values are presented as the average ± standard deviation.

Age Group	65–69	70–74	75–80	>80
CBF Proposed reference value [mL/min]	898.5 ± 119.1	838.5 ± 148.9	805.1 ± 99.3	685.7 ± 112.3
RICA [mL/min]	271.1 ± 63.6	236.0 ± 66.1	234.8 ± 62.3	202.3 ± 38.4
RECA [mL/min]	106.1 ± 35.0	103.7 ± 33.2	94.0 ± 24.14	83.1 ± 36.3
RVA [mL/min]	58.7 ± 29.1	60.2 ± 26.7	62.3 ± 28.4	55.7 ± 24.1
LICA [mL/min]	276.4 ± 57.5	239.8 ± 42.4	245.5 ± 32.3	204.4 ± 47.0
LECA [mL/min]	101.4 ± 30.9	104.7 ± 32.5	89.0 ± 21.9	79.0 ± 33.7
LVA [mL/min]	84.9 ± 33.0	80.4 ± 29.8	70.0 ± 21.5	58.8 ± 13.0

Adapted from Ref. [5]: Copyright Year 2020, Kaszczewski P; Elwertowski, M.; Leszczynski, J.; Ostrowski, T.; Galazka, Z.

**Table 3 diagnostics-12-00766-t003:** Detailed data concerning the flow compensation in different age groups.

Age/Flow Compensation	Significant Compensation	Flow Volume within Reference Value	Decreased Flow Volume
Whole study group	13/46 (28.3%)	20/46 (43.4%)	13/46 (28.3%)
65–69	3/22 (13.6%)	9/22 (41%)	10/22 (45.4%)
70–74	5/15 (33.3%)	9/15 (60%)	1/15 (6.6%)
75–79	0/2 (0%)	0/2 (0%)	2/2 (100%)
>80	5/7 (71.4%)	2/7 (28.5%)	0/7 (0%)

**Table 4 diagnostics-12-00766-t004:** The level of compensation in all extracranial arteries.

Age Group	Artery	Compensation	Reference	*p*-Value < 0.005	Flow Difference	Relative Flow Increase
65–69	ICA	408.9 ± 64.4 mL/min	273.8 ± 60.5 mL/min	yes	135.1 mL/min	149.3%
	ECA	224.6 ± 69.9 mL/min	103.6 ± 32.9 mL/min	yes	121 mL/min	216.9%
	VA	200.3 ± 81.5 mL/min	71.8 ± 32.3 mL/min	yes	128.5 mL/min	279%
70–74	ICA	404.9 ± 67.1 mL/min	237.9 ± 54.3 mL/min	yes	165 mL/min	170.2%
	ECA	212.2 ± 71.5 mL/min	104.2 ± 32.7 mL/min	yes	108 mL/min	203.6%
	VA	173.4 ± 45.4 mL/min	70.3 ± 28.8 mL/min	yes	103.1 mL/min	246.7%
75–79	ICA	excluded	excluded	excluded	excluded	
	ECA	excluded	excluded	excluded	excluded	
	VA	excluded	excluded	excluded	excluded	
≥80	ICA	406 ± 165 mL/min	203.2 ± 42.7 mL/min	yes	202.8 mL/min	199.8%
	ECA	180.25 ± 59.6 mL/min	81 ± 35 mL/min	yes	99.5 mL/min	222.5%
	VA	221.7 ± 81 mL/min	57.3 ± 18.5 mL/min	yes	164.4 mL/min	386.9%

**Table 5 diagnostics-12-00766-t005:** Pathways of collateral circulation in patients with unilateral ICA occlusion.

Age Group	Number of Patients	Occlusion	Contralateral ICA	Contralateral ECA	Contralateral VA	Ipsilateral ECA	Ipsilateral VA
65–69	9	RICA	5/9, 56%	4/9, 44%	6/9, 67%	4/9, 44%	3/9, 33%
	10	LICA	5/10, 50%	3/10, 30%	2/10, 20%	4/10, 40%	8/10, 80%
70–74	8	RICA	7/8, 88%	3/8, 38%	4/8, 50%	5/8, 63%	8/8, 100%
	6	LICA	2/6, 33%	1/6, 17%	2/6, 33%	4/6, 67%	5/6, 83%
75–79	excluded	excluded	excluded	excluded	excluded	excluded	excluded
≥80	7	RICA	4/7, 57%	4/7, 57%	5/7, 71%	5/7, 71%	2/7, 29%
	-	LICA	-	-	-	-	-

**Table 6 diagnostics-12-00766-t006:** Pathways of collateral circulation in patients with bilateral ICA occlusion.

Age Group	Number of Patients	Bilateral Occlusion	LECA	LVA	RECA	RVA
65–69	3	ICA	1/3, 33%	3/3, 100%	3/3, 100%	3/3, 100%
70–74	1	ICA	1/1, 100%	1/1, 100%	1/1, 100%	1/1, 100%
75–79	excluded	excluded	excluded	excluded	excluded	excluded
≥80	-	-	-	-	-	-

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
