# Peer review of "Intracranial Flow Volume Estimation in Patients with Internal Carotid Artery Occlusion"

_diagnostics, 2022, doi:10.3390/diagnostics12030766_

Round 1
Reviewer 1 Report
Thank you for the opportunity to review this manuscript. This is an interesting study but there are several issues that need to be addressed,
Firstly, there are several factors that could have confounded this study which were not addressed such as smoking, Hypertension controlled/uncontrolled, diabetes etc. Also, presence or absence of intracranial stenosis can also affect the flow dynamics. This was not taken into account in this study.
Secondly, ultrasound studies are often user dependent. Was there significant inter-observer variability or has this been established in their prior cohort. This may be worth mentioning in manuscript.
Thirdly, the overall statistical analysis is sound but I would recommend correcting for multiple comparisons as multiple comparisons may lead to high degree of type I errors in analysis.
Lastly, the design of this study is cross-sectional and does not take into account changes in flow dynamics over time after occlusion. It may not be feasible for the authors to do conduct a longitudinal study at this time but his needs to be mentioned in the limitation of the study.
Author Response
Dear Sir or Madam,
On behalf of all authors, I would like to thank You for the time and effort devoted to prepare the review as well as for the constructive comments.
I think that we have already addressed the issues concerning concomitant disorders and inter-observer variability, as clarified in the following paragraphs.
Our team uses the same protocol of the patient’s recruitment in all studies which have been published yet. The study “Intracranial flow volume estimation in patients with Internal Carotid Artery occlusion”, undergoing the review process is the third study of our team, concerning the issue of the assessment of the Cerebral Blood Flow using Doppler Ultrasonography. The previous studies entitled: "Volumetric Carotid Flow Characteristics in Doppler Ultrasonography in Healthy Population Over 65 Years Old” and “Volumetric Flow Assessment in Doppler Ultrasonography in Risk Stratification of Patients with Internal Carotid Stenosis and Occlusion” were published in the Journal of Clinical Medicine in 2020 and 2022, respectively. In all our studies we aimed to exclude the influence of concomitant disorders that could alter the CBF on the results, therefore we created severe inclusion and exclusion criteria, which were described in detail in our first study: "Volumetric Carotid Flow Characteristics in Doppler Ultrasonography in Healthy Population Over 65 Years Old”. The exclusion criteria covered:
“Concomitant diseases: uncontrolled hypertension, ischemic heart disease, heart insufficiency, positive history of heart infraction, positive history of stent implantation to coronary or any other arteries, cardiac arrhythmia, tachycardia, bradycardia, congenital vascular or heart failure, positive history of vascular interventions, presence of endocrine diseases: thyroid goiter, hyper-, hypothyroidism diabetes, adrenal diseases, positive history of thyroid surgery, smoking, alcohol use.”
Kaszczewski, P.; Elwertowski, M.; Leszczynski, J.; Ostrowski, T.; Galazka, Z. Volumetric Carotid Flow Characteristics in Doppler Ultrasonography in Healthy Population Over 65 Years Old. J. Clin. Med. 2020, 9, 1375. https://doi.org/10.3390/jcm9051375
We selected a relatively small group of patients fulfilling the given criteria and briefly described it in the current study:
“Recruitment of the patients to both groups in terms of concomitant disorders was conducted according to the previously described protocol, in order to eliminate its influence on cerebral blood flow volume, except for the ICA stenosis/occlusion [7].”
The influence of the abovementioned diseases on CBF will be addressed in our future research.
The ultrasound is user dependent and always featured with inter-observer variability. Our team aimed to eliminate this issue, therefore all the examinations (also in the already published studies by our team) were conducted by the same sonographer (Piotr Kaszczewski) using the same Canon Aplio i800 ultrasound scanner with Linear i11LX3 transducer (Canon Medical Systems Corporation, Japan).
Following Your suggestion, we implemented the Bonferroni correction in analyzing differences between compensatory increased flow volume in the extracranial arteries and the reference values. As we analyzed 3 arteries in 3 age groups, the significance level was set for (0,05/9 = 0,0055) 0,005. All the comparisons were statistically significant.
The following changes were introduced:
“Additionally, the Bonferroni correction for multiple comparisons was performed in order to reduce the risk of type I errors - the significance level was 0,005”.
The level of significance in the table 4 was changed to 0,005.
Indeed, our study does not take into account changes in flow dynamics over time after occlusion. The measurements were conducted at certain time points. Therefore, this was described in the limitations of the study, the following paragraph was added:
“In our study the CBF values were measured at a certain time – we did not have the possibility to do conduct a longitudinal study, assessing and monitoring the flow dynamics over the time prior to and after the occlusion. This may be an interesting topic for future research.”
All the amendments are highlighted with a green color.
Below I present the list of revisions:
- Abstract: according to the suggestion of the Reviewer 2 the abstract was shortened, and the main outcomes were summarized in less detailed manner. The section results were reformatted to a present form:
- Results: In comparison to CBF values in healthy population 3 subgroups with CBF changes were identified among patients with ICA occlusion: patients with significant volumetric flow compensation (CBF higher than average + standard deviation for healthy equally aged population), patients with flow similar to healthy population (average +/- standard deviation), and patients without compensation (with CBF lower that the average - standard deviation for healthy population). The percentage of patients with significant volumetric flow compensation tend to raise with increasing age, while a simultaneous decline was observed in the group without compensation. The per-centage of patients with flow similar to healthy population remains relatively unchanged. ICA plays the most important role in the volumetric flow compensation in patients with CAO, however the relative increase of the flow in the ICA was smaller than in the ECA and VA. Compensatory increased flow was observed in about 50% of all patent extracranial arteries and was more frequently observed in ipsilateral vessels than in the contralateral ones, both in the ECA and the VA. In patients with CAO there was no decrease of the CBF, ICA, ECA and VA flow volume with in-creasing age
- Introduction: following the suggestion of the Reviewer 3 the information’s concerning anatomic variability of the ICA were mentioned.
- Lines 83-88: the following paragraph was added: “The importance of collateral circulation may be illustrated with the patients with ICA agenesis, aplasia, or hypoplasia. Vast majority of such patients are asymptomatic due to well-developed collateral circulation via either CoW or the persistent embryonical vessels and collaterals. One of the collateral pathways is ECA-ICA anastomosis when the intra-cranial arteries are supplied by the branches of ascending pharyngeal and internal maxillary arteries.
- the new reference was added:
“A.M. Alexandre, E. Visconti, C. Schiarelli, P. Frassanito, A. Pedicelli. Bilateral internal carotid artery segmental agenesis: embryology, common collateral pathways, clinical presentation, and clinical importance of a rare condition. World Neurosurg, 95 (2016)”
- Materials and methods: following the suggestion of the Reviewer 1 the Bonferroni correction for multiple comparisons was performed.
- Lines 172-174: The following sentence was added to the 2.1 Statistical Analysis subsection:
“Additionally, a Bonferroni correction for multiple comparisons was performed in order to reduce the risk of type I errors - the significance level was set to 0,005.
- Results:
- The significance level in the table 4 was changed to <0,005
- According to the suggestion of the Reviewer 2: the equation of the regression was put in corresponding captions for the Figures 5-8.
- Discussion:
- Lines 375-381. Following the suggestion of the Reviewer 3 the following paragraph was added to the discussion:
“The extracranial-intracranial anastomoses covers three main regions: the orbital region (via the ophthalmic artery connecting the internal maxillary and internal carotid vascular beds), the petrous-cavernous region (via the inferolateral trunk, the petrous branches of the internal carotid artery, and the meningohypophyseal trunk to the carotid artery) and the upper cervical region (via the ascending pharyngeal, the occipital, and the ascending and deep cervical arteries to the vertebral artery).”
- Lines 384-386: the following sentence was added: “We have to bear in mind that there are also several other extracranial-intracranial anastomotic pathways, which cannot be assessed in DUS.”
- A new reference was added: S. Geibprasert, S. Pongpech, D. Armstrong and T. Krings. Dangerous Extracranial–Intracranial Anastomoses and Supply to the Cranial Nerves: Vessels the Neurointer-ventionalist Needs to Know. American Journal of Neuroradiology September 2009, 30 (8) 1459-1468; DOI: https://doi.org/10.3174/ajnr.A1500
- Lines 384-386, the following sentence has been added: “We have to bear in mind that there are also several other extracranial-intracranial anastomotic pathways, which cannot be assessed in DUS”.
- Lines 463-466, following the suggestion of the Reviewer 1 the limitation of the lack of the ability to asses the flow dynamics over time was added:
“In our study the CBF values were measured in a certain time – we did not have the possibility to do conduct a longitudinal study, assessing and monitoring the flow dynamics over time prior to and after the occlusion. This may be an interesting topic for future re-search.”
- References: two new references were added. All the references have been updated accordingly in the manuscript body.
- Alexandre AM, Visconti E, Schiarelli C, Frassanito P, Pedicelli A. Bilateral Internal Carotid Artery Segmental Agenesis: Embryology, Common Collateral Pathways, Clinical Presentation, and Clinical Importance of a Rare Condition. World Neurosurg. 2016;95:620.e9-620.e15.
- Geibprasert S, Pongpech S, Armstrong D, Krings T. Dangerous extracranial-intracranial anastomoses and supply to the cranial nerves: vessels the neurointerventionalist needs to know. AJNR Am J Neuroradiol. 2009 Sep;30(8):1459-68.
We hope You will considered the amended version of the manuscript suitable for publication in the Diagnostics.
Faithfully Yours,
Piotr Kaszczewski
Michał Elwertowski
Jerzy Leszczyński
Tomasz Ostrowski
Zbigniew Gałązka

Reviewer 2 Report
The paper by Kaszczewski et al. investigates the correlation between intracranial flow and carotid artery occlusion over a selected set of Patients.
It is well written and the conclusions appear to be relevant in the considered framework.
I agree with its pubblication, provided the Authors asses the following minor issues:
-Reduce the lenght of the abstract and summariz the main outcomes of the paper without entering too much into details.
-Figures 5-8: put the eqaution of the regression line in the corresponding caption;
-The discussion section appears to be difficult to read and it is hard to make a synthesis; I suggest trying to avoid the list of the literature results and to group the observations/comments so as to emphasize in which way your work may add a new insight of the investigated problem.
Author Response
Dear Sir or Madam,
On behalf of all authors, I would like to thank You for the time and effort devoted to preparing the review as well as for the constructive comments concerning its content.
Having received the review, we implemented suggested changes. The abstract has been shortened and the results have been presented in a less detailed manner. The regression equation has been included in the corresponding captions in the Figures 5-8.
You have suggested to avoid the list of the literature results in the discussion, and to group the observations to emphasize our findings. Due to the fact that there is limited data concerning the volumetric flow assessment in extracranial arteries available, especially using Doppler Ultrasonography, our team aimed to write a descriptive discussion. Therefore, we wanted to provide some information concerning all cited works, and to compare it with our own results. We are conscious that it may seem a beat confusing, however, it presents the detailed comparison of our results with other works and describes other factors that may influence the described disease. We would appreciate it if you allowed to preserve the discussion in its unchanged form.
All the amendments are highlighted with a green color.
Below we present the list of revisions:
- Abstract: according to the suggestion of the Reviewer 2 the abstract was shortened, and the main outcomes were summarized in less detailed manner. The section results were reformatted to a present form:
- Results: In comparison to CBF values in healthy population 3 subgroups with CBF changes were identified among patients with ICA occlusion: patients with significant volumetric flow compensation (CBF higher than average + standard deviation for healthy equally aged population), patients with flow similar to healthy population (average +/- standard deviation), and patients without compensation (with CBF lower that the average - standard deviation for healthy population). The percentage of patients with significant volumetric flow compensation tend to raise with increasing age, while a simultaneous decline was observed in the group without compensation. The per-centage of patients with flow similar to healthy population remains relatively unchanged. ICA plays the most important role in the volumetric flow compensation in patients with CAO, however the relative increase of the flow in the ICA was smaller than in the ECA and VA. Compensatory increased flow was observed in about 50% of all patent extracranial arteries and was more frequently observed in ipsilateral vessels than in the contralateral ones, both in the ECA and the VA. In patients with CAO there was no decrease of the CBF, ICA, ECA and VA flow volume with in-creasing age
- Introduction: following the suggestion of the Reviewer 3 the information’s concerning anatomic variability of the ICA were mentioned.
- Lines 83-88: the following paragraph was added: “The importance of collateral circulation may be illustrated with the patients with ICA agenesis, aplasia, or hypoplasia. Vast majority of such patients are asymptomatic due to well-developed collateral circulation via either CoW or the persistent embryonical vessels and collaterals. One of the collateral pathways is ECA-ICA anastomosis when the intra-cranial arteries are supplied by the branches of ascending pharyngeal and internal maxillary arteries.
- the new reference was added:
“A.M. Alexandre, E. Visconti, C. Schiarelli, P. Frassanito, A. Pedicelli. Bilateral internal carotid artery segmental agenesis: embryology, common collateral pathways, clinical presentation, and clinical importance of a rare condition. World Neurosurg, 95 (2016)”
- Materials and methods: following the suggestion of the Reviewer 1 the Bonferroni correction for multiple comparisons was performed.
- Lines 172-174: The following sentence was added to the 2.1 Statistical Analysis subsection:
“Additionally, a Bonferroni correction for multiple comparisons was performed in order to reduce the risk of type I errors - the significance level was set to 0,005.
- Results:
- The significance level in the table 4 was changed to <0,005
- According to the suggestion of the Reviewer 2: the equation of the regression was put in corresponding captions for the Figures 5-8.
- Discussion:
- Lines 375-381. Following the suggestion of the Reviewer 3 the following paragraph was added to the discussion:
“The extracranial-intracranial anastomoses covers three main regions: the orbital region (via the ophthalmic artery connecting the internal maxillary and internal carotid vascular beds), the petrous-cavernous region (via the inferolateral trunk, the petrous branches of the internal carotid artery, and the meningohypophyseal trunk to the carotid artery) and the upper cervical region (via the ascending pharyngeal, the occipital, and the ascending and deep cervical arteries to the vertebral artery).”
- Lines 384-386: the following sentence was added: “We have to bear in mind that there are also several other extracranial-intracranial anastomotic pathways, which cannot be assessed in DUS.”
- A new reference was added: S. Geibprasert, S. Pongpech, D. Armstrong and T. Krings. Dangerous Extracranial–Intracranial Anastomoses and Supply to the Cranial Nerves: Vessels the Neurointer-ventionalist Needs to Know. American Journal of Neuroradiology September 2009, 30 (8) 1459-1468; DOI: https://doi.org/10.3174/ajnr.A1500
- Lines 384-386, the following sentence has been added: “We have to bear in mind that there are also several other extracranial-intracranial anastomotic pathways, which cannot be assessed in DUS”.
- Lines 463-466, following the suggestion of the Reviewer 1 the limitation of the lack of the ability to asses the flow dynamics over time was added:
“In our study the CBF values were measured in a certain time – we did not have the possibility to do conduct a longitudinal study, assessing and monitoring the flow dynamics over time prior to and after the occlusion. This may be an interesting topic for future re-search.”
- References: two new references were added. All the references have been updated accordingly in the manuscript body.
- Alexandre AM, Visconti E, Schiarelli C, Frassanito P, Pedicelli A. Bilateral Internal Carotid Artery Segmental Agenesis: Embryology, Common Collateral Pathways, Clinical Presentation, and Clinical Importance of a Rare Condition. World Neurosurg. 2016;95:620.e9-620.e15.
- Geibprasert S, Pongpech S, Armstrong D, Krings T. Dangerous extracranial-intracranial anastomoses and supply to the cranial nerves: vessels the neurointerventionalist needs to know. AJNR Am J Neuroradiol. 2009 Sep;30(8):1459-68.
We hope You will considered the amended version of the manuscript suitable for publication in the Diagnostics.
Faithfully Yours,
Piotr Kaszczewski
Michał Elwertowski
Jerzy Leszczyński
Tomasz Ostrowski
Zbigniew Gałązka

Reviewer 3 Report
Interesting and well written study regarding Intracranial flow volume estimation in patients with Internal Carotid Artery occlusion.
Justa A few points o incrase the value of your work:
- In introduction you should mention anatomical variability in internal carotid artery (such as ipoplasia and/or aplasia/absence). You can find some interesting descriptions here (Alexandre AM at al. Bilateral Internal Carotid Artery Segmental Agenesis: Embryology, Common Collateral Pathways, Clinical Presentation, and Clinical Importance of a Rare Condition. World Neurosurg. 2016 Nov;95:620.e9-620.e15. doi: 10.1016/j.wneu.2016.08.012. Epub 2016 Aug 13. PMID: 27535626.)
- I would suggest to cite and to comment in discussion the point of extra-intracranial anastomoses (this is the article of reference the the neuroradiology: Geibprasert S, Pongpech S, Armstrong D, Krings T. Dangerous extracranial-intracranial anastomoses and supply to the cranial nerves: vessels the neurointerventionalist needs to know. AJNR Am J Neuroradiol. 2009 Sep;30(8):1459-68. doi: 10.3174/ajnr.A1500. Epub 2009 Mar 11. PMID: 19279274; PMCID: PMC7051597.) The importance of such anastomoses becomes critical in case of ICA occlusion to determine flow compensation.
Author Response
Dear Sir or Madam,
On behalf of all authors, I would like to thank You for the time and effort devoted to preparing the review as well as for the constructive comments concerning its content.
Having received the review, we implemented suggested changes.
All the amendments are highlighted with a green color.
Please find the list of revisions below:
- Abstract: according to the suggestion of the Reviewer 2 the abstract was shortened, and the main outcomes were summarized in less detailed manner. The section results were reformatted to a present form:
- Results: In comparison to CBF values in healthy population 3 subgroups with CBF changes were identified among patients with ICA occlusion: patients with significant volumetric flow compensation (CBF higher than average + standard deviation for healthy equally aged population), patients with flow similar to healthy population (average +/- standard deviation), and patients without compensation (with CBF lower that the average - standard deviation for healthy population). The percentage of patients with significant volumetric flow compensation tend to raise with increasing age, while a simultaneous decline was observed in the group without compensation. The per-centage of patients with flow similar to healthy population remains relatively unchanged. ICA plays the most important role in the volumetric flow compensation in patients with CAO, however the relative increase of the flow in the ICA was smaller than in the ECA and VA. Compensatory increased flow was observed in about 50% of all patent extracranial arteries and was more frequently observed in ipsilateral vessels than in the contralateral ones, both in the ECA and the VA. In patients with CAO there was no decrease of the CBF, ICA, ECA and VA flow volume with in-creasing age
- Introduction: following the suggestion of the Reviewer 3 the information’s concerning anatomic variability of the ICA were mentioned.
- Lines 83-88: the following paragraph was added: “The importance of collateral circulation may be illustrated with the patients with ICA agenesis, aplasia, or hypoplasia. Vast majority of such patients are asymptomatic due to well-developed collateral circulation via either CoW or the persistent embryonical vessels and collaterals. One of the collateral pathways is ECA-ICA anastomosis when the intra-cranial arteries are supplied by the branches of ascending pharyngeal and internal maxillary arteries.
- the new reference was added:
“A.M. Alexandre, E. Visconti, C. Schiarelli, P. Frassanito, A. Pedicelli. Bilateral internal carotid artery segmental agenesis: embryology, common collateral pathways, clinical presentation, and clinical importance of a rare condition. World Neurosurg, 95 (2016)”
- Materials and methods: following the suggestion of the Reviewer 1 the Bonferroni correction for multiple comparisons was performed.
- Lines 172-174: The following sentence was added to the 2.1 Statistical Analysis subsection:
“Additionally, a Bonferroni correction for multiple comparisons was performed in order to reduce the risk of type I errors - the significance level was set to 0,005.
- Results:
- The significance level in the table 4 was changed to <0,005
- According to the suggestion of the Reviewer 2: the equation of the regression was put in corresponding captions for the Figures 5-8.
- Discussion:
- Lines 375-381. Following the suggestion of the Reviewer 3 the following paragraph was added to the discussion:
“The extracranial-intracranial anastomoses covers three main regions: the orbital region (via the ophthalmic artery connecting the internal maxillary and internal carotid vascular beds), the petrous-cavernous region (via the inferolateral trunk, the petrous branches of the internal carotid artery, and the meningohypophyseal trunk to the carotid artery) and the upper cervical region (via the ascending pharyngeal, the occipital, and the ascending and deep cervical arteries to the vertebral artery).”
- Lines 384-386: the following sentence was added: “We have to bear in mind that there are also several other extracranial-intracranial anastomotic pathways, which cannot be assessed in DUS.”
- A new reference was added: S. Geibprasert, S. Pongpech, D. Armstrong and T. Krings. Dangerous Extracranial–Intracranial Anastomoses and Supply to the Cranial Nerves: Vessels the Neurointer-ventionalist Needs to Know. American Journal of Neuroradiology September 2009, 30 (8) 1459-1468; DOI: https://doi.org/10.3174/ajnr.A1500
- Lines 384-386, the following sentence has been added: “We have to bear in mind that there are also several other extracranial-intracranial anastomotic pathways, which cannot be assessed in DUS”.
- Lines 463-466, following the suggestion of the Reviewer 1 the limitation of the lack of the ability to asses the flow dynamics over time was added:
“In our study the CBF values were measured in a certain time – we did not have the possibility to do conduct a longitudinal study, assessing and monitoring the flow dynamics over time prior to and after the occlusion. This may be an interesting topic for future re-search.”
- References: two new references were added. All the references have been updated accordingly in the manuscript body.
- Alexandre AM, Visconti E, Schiarelli C, Frassanito P, Pedicelli A. Bilateral Internal Carotid Artery Segmental Agenesis: Embryology, Common Collateral Pathways, Clinical Presentation, and Clinical Importance of a Rare Condition. World Neurosurg. 2016;95:620.e9-620.e15.
- Geibprasert S, Pongpech S, Armstrong D, Krings T. Dangerous extracranial-intracranial anastomoses and supply to the cranial nerves: vessels the neurointerventionalist needs to know. AJNR Am J Neuroradiol. 2009 Sep;30(8):1459-68.
We hope You will considered the amended version of the manuscript suitable for publication in the Diagnostics.
Faithfully Yours,
Piotr Kaszczewski
Michał Elwertowski
Jerzy Leszczyński
Tomasz Ostrowski
Zbigniew Gałązka

Round 2
Reviewer 1 Report
Thank you for your response. All issues have been appropriately addressed.
This manuscript is a resubmission of an earlier submission. The following is a list of the peer review reports and author responses from that submission.
Round 1
Reviewer 1 Report
This study included unilateral or bilateral ICA occlusion,using Doppler ultrasound (DUS) examination measuring blood flow volume in the internal carotid artery (ICA), external carotid artery (ECA), and vertebral artery (VA) to evaluate the flow compensation.The author found ICA is the most important vessel in volumetric flow compensation. However, the accentuated role of the ECA and VA highlights their impact which might be more important .No physiological correlations between flow volume and the increasing age were observed in patients with CAO.
Comments:
- This manuscript needs to be improved the language logic and readability.
- The abstract section needs to be rewritten to present the result more clearly.
- Please explain the “reference values” more clearly.
- The Result section of the manuscript needs to be rewritten to make the reader more clearer to understand the statistical results.
- A brief summary of the previous study in Discussion section, it needs to be shorted and language simplified.
6) As the author said patients with ICA occlusion, although in the ICA the flow volume can increase up, The ability of VA to increase flow volume is surprisingly high. Hower, it is not able to determine the correlation of flow compensation with the anatomy of intracranial arteries. The authors need to further elaborate whether there is any intracranial vascular stenosis in the enrolled population
Author Response
Dir Sir or Madam,
On behalf of all authors I would like to thank You for the time and effort devoted to prepare the review as well as for the constructive comments, which provided valuable insights, indispensable in refining manuscript content.
Having received the review, we put all our efforts to implement necessary changes and prepare the manuscript body in accordance with Your suggestions. We hope You will find the improved version of manuscript presenting results in more comprehensible manner. I would like to ensure Your Party that all minor issues were thoroughly corrected, according to the review.
Please find the detailed list of amendments below:
1: Abstract: the section “Results” has been rewritten to present the results of the manuscript more clearly:
“Results: The percentage of patients with significant volumetric flow compensation tends to raise with increasing age, while a simultaneous decline was observed in the group without compensation. ICA plays the most important role in the volumetric flow compensation in patients with CAO, however the relative increase of the flow in the ICA (1,5-2x) was smaller than in the ECA (2,0-2,2x) and VA (2,5-3,2x). Flow compensation has multivessel character – compensatory increased flow was observed in about 50% of all patent extracranial arteries (ICA, ECA and VA). Flow compensation was more frequently observed in ipsilateral vessels than in the contralateral ones (23 vs 16 – ECA, and 22 vs 17 – VA). In patients with CAO there was no decrease of the CBF, ICA, ECA and VA flow volume with increasing age.”
2: Introduction: The introduction has been rewritten. First 5 paragraphs from the discussion has been included into introduction.
3.Materials and Methods:
- Table 2 – headings in the table 2 has been modified – “CBF” and “years old” have been added
- The terms: significant compensation, flow volume within reference values and no compensation have been defined:
“In this study: the values exceeding the proposed reference value: average + standard deviation are referred as to “compensatory increased flow” or “significant compensation”
The values within proposed reference are referred to as “Flow volume within reference value” – in the presence of major reduction of flow in one of carotid arteries, the increase in the other vessels allows to maintain the CBF within proposed standards.
The blood flow volume, which was lower than proposed reference value: average - standard deviation, are referred as to “no compensation” or “decreased flow volume”.”
- Results: The section results has been modified. Several figures has been prepared to present the results more clearly. We do hope that our amendments will improve the clarity of the paragraph. However, if the Reviewer will find it in a different way, we kindly please for suggestions what and how to improve.
- Figure 1 – presenting graphically the changes in the percentage of patients with flow compensation, with flow similar to reference values and without compensation has been added: “Figure 1. Tendencies in CBF changes in different age groups. The percentage of patients with signifficant flow com-pensation tend to raise with incrasing age. The opposite trend is observed in group with dicreased flow volume. Grup 75-79 was excluded from the analysis.”
- Figure 2 – presenting the percentage of the ICA, ECA and VA with significant flow compensation has been added: “Figure 2. The percentage of the patent arteries with and without flow compensation. The compensatory in-creased flow volume was most frequently observed in the contralateral ICA – 54,76%, in 48,91% of the ECAs, and in the 51,09% of the VAs. “
- Sentence: The compensatory increased flow was more frequently observed in ipsilateral vessels than in the contralateral ones – see figure 3.
- Figure 3 presenting the differences in flow compensation between ipsilateral and contralateral vessels (ECA and VA) has been added: “Figure 3. Differences in the compensation between ipsilateral and contralateral vessels. The compensatory increased flow was more frequently observed in the ipsilateral vessels (23 vs 16 – ECA and 22 vs 17 - VA)”
- Paragraph 3.5 and figure 8 presenting the changes in the number of females and males in the study groups has been added:
“In our study group we observed continuous decrease in percentage of female patients. No female patients were observed in the oldest ≥ 80 years old age group – see figure 8.”
Figure 8. Comparison of the percentage of female and male in our study group
- Discussion:
- The discussion paragraph has been modified. First five paragraphs have been included introduction.
- Several paragraphs have been added in order to discuss our results more thoroughly”
- Lines 308-313: If we look at the percentage – the volumetric flow compensation was present in more or less 50% of all patent ICAs (most frequently 54,8%), ECAs (48,9%) and VAs (51,1%). However the number of the patent ECAs and VAs is two times larger than the number of patent ICAs, which stresses the important role of these vessels as the pathways of collateral circulation.
- Line 333: …”, in the group of healthy volunteers” has been added.
- Lines 338 – 341: “In patients with carotid artery occlusion we didn’t observe tendencies and trends featuring healthy population. In patients with CAO, a CBF remains almost unchanged with increasing age. A slight decreasing tendencies in a flow volume were observed in the ECA and VA, while increasing trend was noted in ICA.” has been added.
- Lines 353-359 have been added: “In our study a one surprising observation has been made: the percentage of female patients with CAO decreased much more rapidly than in males. We do not want to draw any conclusion basing on that observation, however this astonishing tendency has to be verified in a studies on a much larger population. This observation may be biased with not representative population, or to small sample volume. We analyzed the patients being under supervision of our department, which is representative of a regional population, and observed such tendency. “
- Lines 367-374 has been added: “Another shortcoming is the fact that it is a single center observational cohort study, which is conducted on a relatively small number of patients, not allowing for a subgroup analysis and the reference values comes from our previous research – these are single center observations, which means published, but not validated data. In order to introduce this method as diagnostic tool the randomized, multi-center studies on a representative groups of patients have to be conducted.”
- Lines 390-397 has been added: “Conditions of intracerebral circulation in patients with severe (≥70%) ICA narrowing are subject to different flow compensation with degree of flow volume loss varying even with similar degree of stenosis. ICA occlusion provides excellent pathophysiological model of flow impairment which is not affected by originally disturbed vessel thus accentuating other mechanism providing proper brain perfusion. In our study contralateral ICA was most effective single vessel supplying surplus flow to underperfused regions of brain. However combined additional flow in ECA + VA was more effective as provider of supplementary brain perfusion.”
- Conclusions:
- Conclusion 2 has been modified and rewritten in a more carefull and conservative way: “In the study group, the growing tendency in the percentage of patients with significant volumetric flow compensation was observed with increasing age, while the opposite trend (decreasing percentage of patients) was observed in the group without flow compensation.”
I hope you will find the revised version of manuscript suitable for publication in Diagnostics.
Faithfully Yours,
Piotr Kaszczewski
Reviewer 2 Report
Authors have written a good manuscript focused in an important topic but with several things that must be improved.
Abstract:
- What does “significant, massive ischemic” mean?
Introduction
- Introduction should be focused in the explanation of the problem that the study will try to solve or study.
Methods
- Flow volumes have been evaluated have been stablished using a published but not-validated data.
- Sample size is very low.
- What are the definition of significant compensation, Flow volume within reference value and decreased flow volume?
Results
- Results are expressed too schematically without a clear nexus.
Discussion/Conclusion:
- First 5 paragraphs’ information should be included in the introduction.
- Data from the manuscript are not well discussed, discussion is too focused in previous literature.
- Discussion is too long
- Due to the low number of patients and the discrepancy with your previous own data authors don’t have data which support the sentence: “The observation of the increasing number of patients with flow compensation with increasing age suggest, that compensations promotes patients survival”
English should be reviewed
Author Response

(The authors gave the same response as above.)

Round 2
Reviewer 1 Report
This study included unilateral or bilateral ICA occlusion,using Doppler ultrasound (DUS) examination measuring blood flow volume in the internal carotid artery (ICA), external carotid artery (ECA), and vertebral artery (VA) to evaluate the flow compensation. The author found ICA is the most important single vessel in volumetric flow compensation. However, the accentuated role of the ECA and VA highlights their impact which might be more important. No physiological correlations between flow volume and the increasing age were observed in patients with CAO.
Comments:
This manuscript needs to be improved the language logic and readability.
The abstract section needs to be rewritten to present the result more clearly.
Please explain the “reference values” more clearly.
The Result section of the manuscript needs to be rewritten to make the reader more clearer to understand the statistical results.
A summary of the previous study in the Discussion section needs to be shorted and the language simplified.
6) As the author said patients with ICA occlusion, although in the ICA the flow volume can increase up, The ability of VA to increase flow volume is surprisingly high. However, it is not able to determine the correlation of flow compensation with the anatomy of intracranial arteries. The authors need to further elaborate on whether there is any intracranial vascular stenosis in the enrolled population.
Reviewer 2 Report
Introduction
- Introduction section is too long and it is not focused on the topic of the article.
- How can you relate “the presence of sufficient and well-developed collateral circulation” with flow volume measured by ultrasound?
Methods
- CBF is not defined
- When you say extracranial arteries in patients with ICA occlusion, does it mean external CA or common CA? Or contralateral ICA?
Results
- Results section are too schematic
- Format of figures must be improved
- There are any relation between patrons of flow compensation and outcomes?
Discussion
- Probably flow compensation measured by ultrasound is an image of Willis polygon pattency not about collaterals. Have you checked the collateral status of these patients?
- If collateral status has not been checked, discussion should be refocused in Willis polygon patency.